# Assessing modern river sediment discharge to the ocean using satellite gravimetry

Maxime Mouyen [1], Laurent Longuevergne [1], Philippe Steer [1], Alain Crave[1], Jean-Michel Lemoine[2], Himanshu Save[3] & Cécile Robin[1]

Recent acceleration of sand extraction for anthropic use threatens the sustainability of this major resource. However, continental erosion and river transport, which produce sand and sediment in general, lack quantification at the global scale. Here, we develop a new geodetic method to infer the sediment discharge to ocean of the world's largest rivers. It combines the spatial distribution of modern sedimentation zones with new high-resolution (~170 km) data from the Gravity Recovery and Climate Experiment (GRACE) mission launched in 2002. We obtain sediment discharges consistent with in situ measurements for the Amazon, Ganges-Brahmaputra, Changjiang, Indus, and Magdalena rivers. This new approach enables to quantitatively monitor the contemporary erosion of continental basins drained by rivers with large sediment discharges and paves the way toward a better understanding of how natural and anthropic changes influence landscape dynamics.

[1] Univ Rennes, CNRS, Géosciences Rennes, UMR 6118, Rennes 35000, France. [2] CNES/GRGS, 18 Avenue E. Belin, Toulouse Cedex 31401, France. [3] Center for Space Research, The University of Texas at Austin, Austin 78712, USA. Correspondence and requests for materials should be addressed to M.M. (email: maxime.mouyen@univ-rennes1.fr)

Continental erosion is originally controlled by climate[1,2] and tectonics[3,4], but can be enhanced by human activities such as large-scale agriculture[5], deforestation[6], and sand extraction[7]. Erosion limits the growth of mountain belts and produces sediment that is mostly evacuated by rivers. Downstream, the supply of sediment shapes deltas and carries organic matter and nutrients to the ocean, which are fundamental for marine ecosystems[8] and for oil and gas reservoirs[9]. However, the global sediment discharge to the ocean remains unknown and its measurement still represents a daunting challenge in Earth sciences. Indeed, it requires continuous measurements of suspended and bedload sediment transport at the mouth of every river, which is difficult and expensive. Also, these measurements should rely on homogenous measuring techniques. Such a demanding survey cannot be achieved as even modern in situ measurements of sediment discharge inherently lead to spatially sparse data, specific to one location along one river. In addition, such datasets are also temporally discontinuous, recorded at a specific date or with a frequency that rarely accounts for extreme events and the temporal variability of river discharge. To finish, these datasets are incomplete and biased, as bedload transport is rarely monitored[10]. Nevertheless, basin-scale eroded sediment ultimately concentrates in oceanic size-limited sedimentation zones offshore river mouths, where they must provide a clear gravimetric signal. For instance, an accumulation of 0.5 cm year$^{-1}$ of sediment replacing water over a 200-km radius region leads to a ~1 Gt year$^{-1}$ net mass increase. Here, we demonstrate that the associated increment in gravitational attraction is measurable by the GRACE (Gravity Recovery and Climate Experiment[11]) satellite mission. Our main hypothesis is that the mass of sediment exiting a river and accumulating in the ocean over a given time $t$ is the sediment discharge of that river multiplied by $t$. GRACE measures gravity changes as a function of time and position over the whole Earth, at monthly time scales. Gravity being physically linked to masses, GRACE data is fundamentally sensitive to the amplitude of mass variations, with no minimum threshold of spatial extension[12]. GRACE can thus overcome the aforementioned three limitations of in situ measurements and provide critical new insights into sediment discharge from major rivers.

## Results

**Advection model.** GRACE, as the first gravimetric satellite of its generation, proved to be successful in quantifying mass transfers due to various processes, including ice cap and glacier melting[13], terrestrial water storage variations[14,15], or deep Earth mass redistributions resulting from earthquakes[16]. While fluid mass transport and large earthquakes redistribute tens to hundreds of gigatons (Gt) of material per year, the highest river sediment discharges probably reach ~1 Gt year$^{-1}$, as in the case of the Amazon and the Ganges-Brahmaputra delta[10]. In addition, earthquakes, ice, and hydrological mass changes can be identified by alternative methods, but the location of modern sedimentation zones lacks the robust constraints required to drive GRACE interpretation[17]. Submarine fans of large rivers have been extensively studied using coring and seismic reflection techniques. However, they integrate sedimentation processes occurring over hundreds to millions of years[18], whereas GRACE focuses on sedimentation zones at the decadal time scale (the GRACE mission lasted from 2002 to 2017). Therefore, we develop a numerical model of oceanic sediment transport to estimate contemporary sedimentation zones for 13 rivers with the world's highest sediment discharges (see Fig. 1 and Supplementary Table 1). A schematic summary of the entire workflow of our study is given in Supplementary Fig. 1.

The advection model combines the monthly sediment discharge of each river (see Supplementary Fig. 2) with time-variable oceanic 3D current velocities produced by the project Estimating the Circulation and Climate of the Ocean, Phase II (ECCO2[19], see Methods and Supplementary Table 3), of which the horizontal resolution is 0.25°, vertical resolution ranges from 10 to 450 m (resolution decreases with depth), and temporal resolution is 3 days. This model tracks sinking sediment particles in accordance with the dynamics of both the sediment discharge and the currents. The resulting sedimentation zones (Fig. 2a–c and Supplementary Fig. 3) have the same spatial resolution as ECCO2 (0.25°) and contain the total amount of sediment that was delivered at their respective rivers' outlets over the time of simulation (10 years). These sedimentation zones are primarily controlled by two factors. The first one is the velocity of the oceanic currents (horizontal and vertical directions), which transport suspended sediment particles away from river mouths. As oceanic currents vary over the year, the accurate definition of sediment accumulation patterns requires the relative temporal distribution of river sediment discharge. This relative distribution is taken from the temporal variations of in situ suspended sediment discharge, defined at the monthly scale

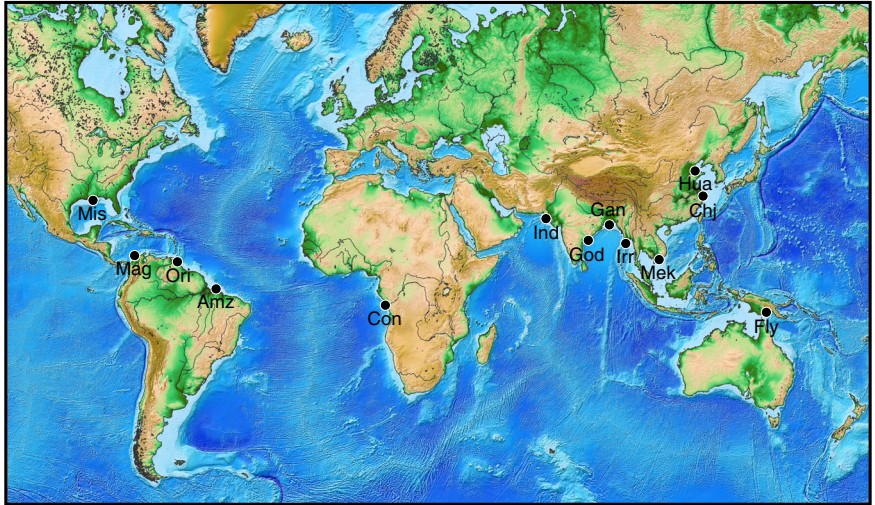

**Fig. 1** Rivers which sediment discharge is estimated in this work. Gan: Ganges-Brahmaputra, Irr: Irrawaddy, Amz: Amazon, Chj: Changjiang, Ind: Indus, Mag: Magdalena, God: Godavari, Mek: Mekong, Con: Congo, Hua: Huanghe, Mis: Mississippi, Ori: Orinoco. Figure 1 created using GMT software[68]

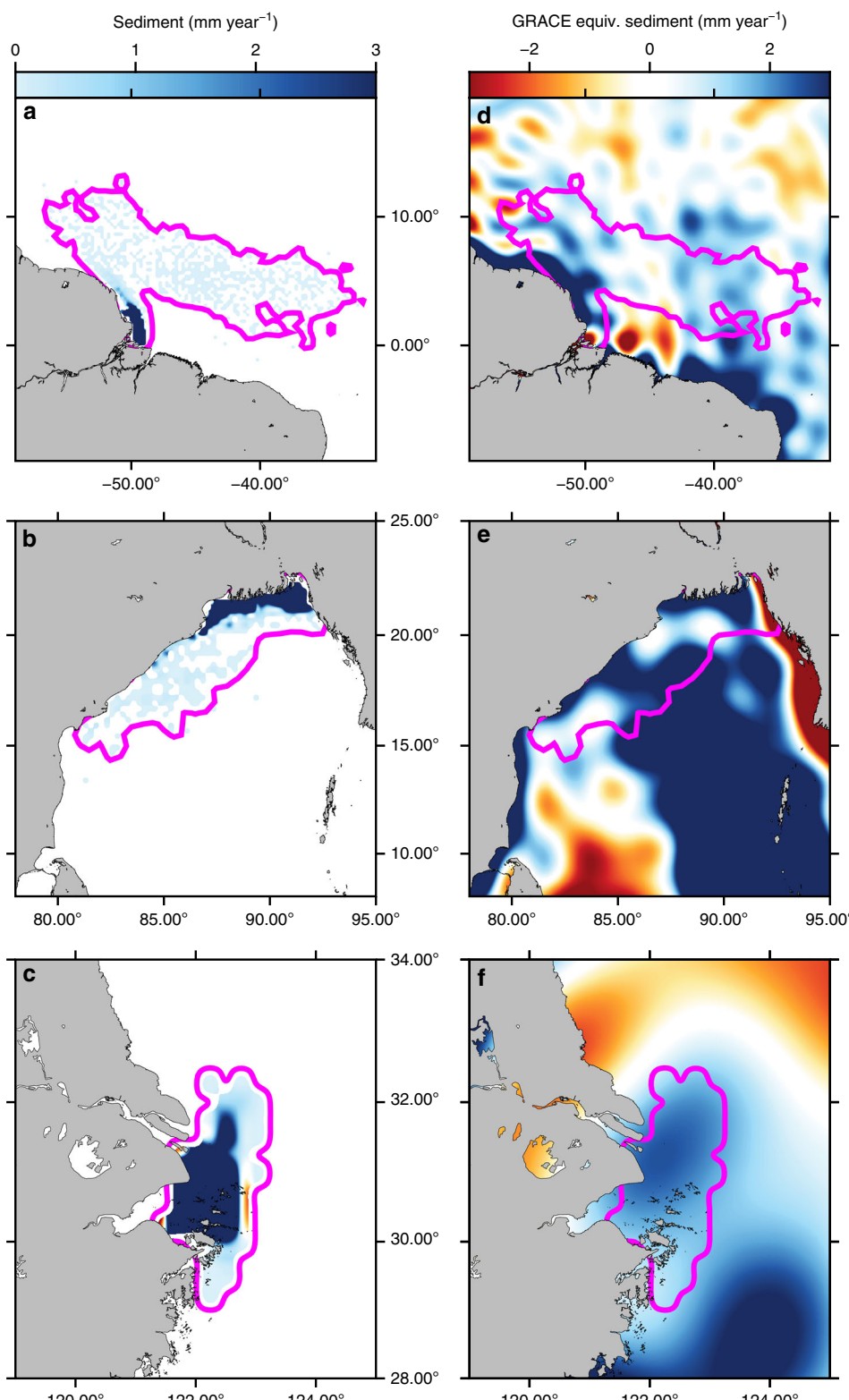

**Fig. 2** Maps of annual offshore sedimentation and collocated GRACE signal. Modeled sedimentation rates out of **a** the Amazon, **b** the Ganges-Brahmaputra, and **c** the Changjiang (or Yangtze) for 10-μm diameter sediment particles. The sedimentation area contour is the pink line. **d**–**f** GRACE equivalent sediment thickness (assuming that sediment replaces water) over the same areas, the sedimentation area contour is also superimposed. Mass variations over the continents are masked to improve figure readability (see Supplementary Fig. 4 for unmasked GRACE signal over the continents). Figure 2 created using GMT software[68]

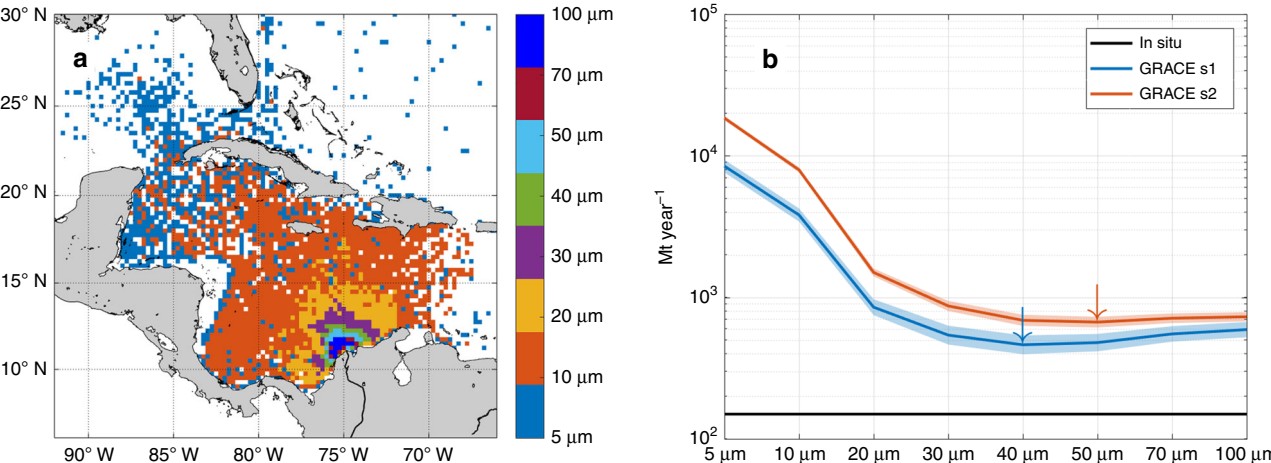

**Fig. 3** Example of the grain size control of the modeled sediment distribution at the Magdalena river. **a** Sedimentation area decreases when grain size increases due to a faster sink relative to the horizontal advection capability of the sea currents. **b** Influence of the grain size on the SSD. For the Magdalena, SSD is closer to the in situ sediment discharges when grain size is around 40–50 μm (see blue and red arrows for s1 and s2 solutions, respectively) and in any case larger than 40 μm. Results at the other 12 rivers are in Supplementary Fig. 6. The hue around each GRACE solution represents the uncertainty derived from the signal (sediment) to noise (GRACE data) analysis (see Methods)

(Supplementary Fig. 2). The second factor is the settling velocity of the particles, which is a function of their density and size. The density is kept constant ($2500 \, \mathrm{kg \, m^{-3}}$) but reasonable grain sizes ranging from 5 to 100 μm (silt to very fine sand) are tested for each studied river.

The resulting distribution emphasizes areas with larger relative sedimentation rates localized near each river mouth, which are more likely to influence GRACE observations (Fig. 2d–f, Supplementary Fig. 3). The modeled amount of deposited sediment is eventually normalized by the total mass of sediment delivered by each river. This leads to the spatial distributions of offshore sedimentation independent of estimated in situ total sediment discharge. The mass can now be estimated from the GRACE data in this sedimentation zone.

**GRACE satellite-derived sediment discharge (SSD).** We analyze the linear gravity change computed from 10 years (2002–2011) of monthly regularized GRACE solutions. They are originally provided in equivalent water thickness that we convert to equivalent sediment thickness assuming that sediment (density 2.5) replaces water (density 1). This distribution is spatially filtered as GRACE (Methods) based on the full spectral bandwidth available in regularized GRACE solutions[20]. These spherical harmonic solutions are computed up to degree and order 120 (see Methods), which corresponds to a spatial resolution of ~170 km. Note that GRACE is sensitive to masses, not spatial scales[12]. The 170-km resolution should be understood as an integration domain rather than a pixel-size information. These offer higher-resolution solutions with improved signal-to-noise ratio (SNR) as compared to standard solutions (truncation at 96°, while further smoothing is required to remove noise, constraining effective resolution <500 km). Such high resolution is required to recover focused sediment mass changes. More details on the quality of regularized GRACE solutions with respect to usual solutions, their ability to estimate localized mass changes near the coast and separate from continental hydrological signals can be found in the Methods. Although targeted sediment accumulation zones are located in the oceanic domain, continental hydrology remains a major contributor to GRACE measurements because of leakage[21]. We test two different strategies to correct this leakage, either based on a hydrological model ensemble (s1) or on a hydrological model assimilating GRACE (s2) (see Methods and

Supplementary Fig. 5). GRACE SSD is then obtained by fitting in a least-square sense the modeled distribution of sediment to the distribution of GRACE linear trends over 2002/2004 and 2012/2012 for both s1 and s2 and with uncertainty estimates based on the SNR (see Methods). For the Ganges-Brahmaputra, Godavari, Irrawaddy, and Mekong rivers, the analyzed time range is limited to May 2005–2012, to avoid the co-seismic contributions of the Sumatra and Nias earthquakes (26 December 2004: $M_\mathrm{w} = 9.3$ and 28 March 2005: $M_\mathrm{w} = 8.7$, respectively) in their sedimentation areas[22].

**Validation of SSD.** A wide range of SSDs can be obtained depending on the chosen grain size and s1/s2 processing strategy. We consider that a SSD is valid if (1) s1 and s2 hydrological corrections provide similar SSD values with similar grain sizes and (2) the most probable SSD is expected to reduce the difference with the in situ estimates (Fig. 3 and Supplementary Fig. 6). Given their acknowledged uncertainty, in situ estimates are considered as first-order constraints rather than accurate values. Under these conditions, valid SSD are proposed for 4 out of the 13 studied rivers, including the Amazon (s1: $560 \pm 90 \, \mathrm{Mt \, year^{-1}}$; s2: $610 \pm 170 \, \mathrm{Mt \, year^{-1}}$), Changjiang (s1: $430 \pm 100 \, \mathrm{Mt \, year^{-1}}$; s2: $330 \pm 100 \, \mathrm{Mt \, year^{-1}}$), Indus (s1: $150 \pm 70 \, \mathrm{Mt \, year^{-1}}$; s2: $350 \pm 120 \, \mathrm{Mt \, year^{-1}}$), and Magdalena (s1: $460 \pm 90 \, \mathrm{Mt \, year^{-1}}$; s2: $670 \pm 90 \, \mathrm{Mt \, year^{-1}}$) rivers (Fig. 4 and Table 1). Indeed, these rivers have the highest sediment discharges in the world, hence the highest contribution to the GRACE data. Further, for the Changjiang and Magdalena, grain sizes for each processing strategy are identical, supporting the robustness of the SSD estimates. This demonstrates the potential of this novel approach to monitor the rate of sediment mass accumulation at the mouth of rivers having large sediment discharges.

For the Ganges-Brahmaputra, even if the SSD fails to fulfill the imposed conditions, we note that SSD obtained with s1 is in excellent agreement with in situ values and is the highest SSD value measured worldwide (s1: $1140 \pm 200 \, \mathrm{Mt \, year^{-1}}$). For the Irrawaddy, removing the coseismic effect of Sumatra-Nias earthquakes is not a sufficient correction because of ongoing high-amplitude post-seismic effects, which build large gravity trends and eventually hinder the potential contribution of sediment accumulation (Supplementary Fig. 8 and Supplementary Table 2). Godavari and Mekong river's SSD are also

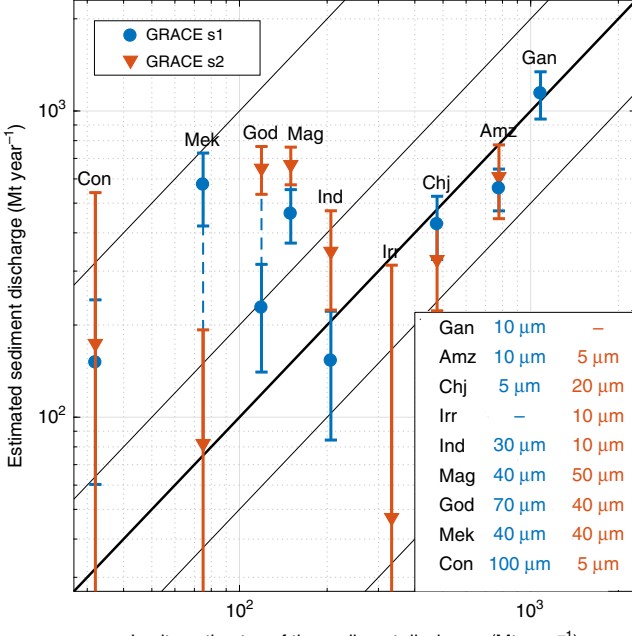

**Fig. 4** GRACE satellite-derived sediment discharge (SSD) with standard deviation vs. in situ suspended sediment discharge. For clarity, we only show positive values of SSD for grain sizes (bottom right subpanel) that better compare with in situ sediment discharge for each hydrological correction (s1 and s2). Ganges-Brahmaputra (Gan) and Irrawaddy (Irr) have negative SSD for s2 and s1 solutions, respectively. Amz: Amazon, Chj: Changjiang, Ind: Indus, Mag: Magdalena, God: Godavari, Mek: Mekong, Con: Congo

significantly modified by removing Sumatra-Nias' coseismic effect (Supplementary Table 2) and brought closer to in situ estimates, yet s1 and s2 solutions are not similar (factor 3 and 7 for Godavari and Mekong, respectively). Estimates for the Congo river are not valid since s1 and s2 lead to very different grain sizes, around 100 and 5 μm, and to very different sedimentation areas, 5300 and 570,000 km², respectively.

For rivers with smaller sediment discharges, other processes leading to gravity changes can dominate mass accumulation by sedimentation. In particular, we find negative SSD for the Mississippi, Orinoco, Fly, and Huanghe rivers (Table 1). The Mississippi sedimentation zone experiences subsidence possibly related to significant mass transfers (e.g., oil extraction) and visco-elastic deformation[23–26]. The Orinoco, Huanghe, and the Fly rivers deliver sediment in shallow bays surrounded by continents (and island arcs for Orinoco) inducing significant continental leakage and inaccurate oceanic corrections. The negative SSD for the s2 correction for the Ganges-Brahmaputra river probably results from the difficulty of hydrological models to account for surface-water and ground-water storage in this region[27].

## Discussion

Our approach brings new perspectives for the monitoring of continental erosion and rates of sediment mass accumulation out of rivers having the world's largest sediment discharges using satellite gravimetry. This also highlights the wealth of GRACE observations to capture various Earth surface processes. Compared to classical in situ measurements, that directly measure suspended sediment fluxes in rivers with the limitation of local and discontinuous sampling, this novel method measures

sediment discharges derived from continuous mass accumulation in the ocean. Such satellite-derived estimation requires determining the extent of modern sedimentation zones, which is performed here using a sediment advection model. With the exception of local studies[28], there is a dire lack of data documenting modern sedimentation zone and especially the grain size distribution for rivers at the global scale. Yet the grain size significantly controls the extension of the estimated sedimentation zone. We thus tested a range of possible grain sizes, but robust estimates of the grain size distribution compiled in accessible database would make our sedimentation zones estimates more reliable. Another limitation comes from the GRACE data, which inherently suffer from leakage of hydrological signals into the coastal ocean[21]. Although such a leakage effect is efficiently reduced in the regularized solutions we use (see Methods), it prevents from identifying masses of sediment accumulated out of rivers with sediment discharges smaller than ~150 Mt year⁻¹ (Fig. 4 and Table 1) and makes this study focused on rivers having the world's largest sediment discharges.

As an integrated sediment mass estimates, GRACE data complement in situ measurements by widening the range of spatial and temporal scales and providing a homogenous method to quantify these masses at the global scale. Therefore, it is expected that GRACE-type satellite missions would deliver new insights into processes and mechanisms controlling sediment redistribution. Global sediment erosion and transport models[29] would benefit from this novel sediment mass observation, independent from rivers' flow information[30].

Since sediment particles are continuously exported to the ocean, the total mass of the sedimentation zones and, therefore, the SNR in the gravity time series keeps increasing. As a result, longer gravity time series recorded by GRACE Follow-on missions[31] together with continuous improvement of GRACE data quality will make SSD estimations more accurate. This could also reveal pluri-annual or longer-term variations of sediment discharge potentially induced by climate change. At this stage, GRACE data does not allow to detect monthly or annual variations of sediment discharge. GRACE Follow-on and future developments could unlock this limitation and resolve the short-timescales dramatic variations of sediment discharge following earthquakes, typhoons, or El Niño events[32,33]. Overall, satellite gravimetry represents a timely opportunity to quantify the impact of natural processes and anthropic changes, including deforestation, land use change, river damming, and sand extraction, on sediment fluxes from the continent to the ocean.

## Methods

**In situ sediment discharge.** Using published works, we first compile the monthly sediment discharge of each river considered in this study (Supplementary Fig. 2 and Supplementary Table 1): Amazon[34], Ganges-Brahmaputra[35], Changjiang[36], Irrawaddy[37], Indus[38], Huanghe[39,40], Magdalena[32], Godavari[41], Fly[42,43], Orinoco[44], Mekong[45,46], Mississippi[47], Congo[44]. Temporal variability in sediment export (monthly relative fraction) is required for the sediment advection model (described in the next paragraph) to model the fraction of the total yearly sediment discharge that is exported each month. The yearly sediment export is conserved as a reference for comparison with the SSD, once the latter is computed.

**Sediment advection model.** We assume that the particle is advected at the same velocity as the sea current velocity. While experiencing suspended transport, the sediment particle also sinks considering the Stokes settling velocity (Supplementary Fig. 9) due to gravity at a velocity $V_s = g(\rho_p - \rho_w)D_p^2/18\mu$, where $g = 9.8$ m s⁻² is the gravity at the Earth's surface, $\rho_p = 2500$ kg m⁻³ and $\rho_w = 1030$ kg m⁻³ are the volume masses of the sediment particle and the seawater, respectively. $D_p$ is the particle diameter and $\mu = 2 \times 10^{-3}$ Pa s is the seawater viscosity. Grain size controls significantly the ability of the particle to be transported and deposited far from the river mouth. We run the advection model using different grain sizes: 5, 10, 20, 30, 40, 50, 70, and 100 × 10⁻⁶ m. In situ, several grain sizes may be identified. For instance, at the mouth of the Amazon river, two main modes of grain size (10 and 100 μm) exist, and their relative proportions change across the year. Although our

**Table 1 Satellite-derived sediment discharge (SSD) summary**

| River | In situ | SSD | Grain size | Grain size range | Range of the sedimentation zone area | SNR in GRACE data |
|---|---|---|---|---|---|---|
| | Mt year⁻¹ | Mt year⁻¹ | µm | µm | ×1000 km² | |
| Amazon | 778 | **560 ± 90** | **10** | **10** | **630** | **2.4** |
| | | *610 ± 165* | *5* | *5* | *3000* | *0.8* |
| Ganges-Brahmaputra | 1081 | **1140 ± 200** | **10** | **10** | **210** | **1.2** |
| | | *−890 ± 750* | *–* | *–* | *–* | *0.1* |
| Changjiang | 477 | **430 ± 100** | **5** | **5** | **110** | **1.3** |
| | | *330 ± 100* | *20* | *10–100* | *40–12* | *1.0* |
| Mississippi | 51 | **−130 ± 310** | **–** | **–** | **–** | **0.1** |
| | | *−190 ± 380* | *–* | *–* | *–* | *0.1* |
| Irrawaddy | 333 | **−1100 ± 610** | **–** | **–** | **–** | **0.5** |
| | | *50 ± 270* | *10* | *10* | *220* | *0.3* |
| Indus | 206 | **150 ± 70** | **30** | **30–100** | **15–10** | **0.6** |
| | | *350 ± 120* | *10* | *10* | *80* | *1.5* |
| Orinoco | 76 | **−250 ± 450** | **–** | **–** | **–** | **0.2** |
| | | *−110 ± 240* | *–* | *–* | *–* | *0.1* |
| Godavari | 119 | **230 ± 90** | **70** | **70–100** | **24–12** | **3.0** |
| | | *650 ± 120* | *40* | *40–50* | *71–43* | *5.8* |
| Mekong | 75 | **580 ± 150** | **40** | **20–100** | **12–8** | **0.4** |
| | | *80 ± 110* | *40* | *40–100* | *8* | *0.4* |
| Magdalena | 150 | **460 ± 90** | **40** | **40–50** | **50–30** | **2.3** |
| | | *670 ± 90* | *50* | *40–70* | *50–16* | *4.8* |
| Fly | 111 | **−1980 ± 660** | **–** | **–** | **–** | **0.9** |
| | | *−570 ± 470* | *–* | *–* | *–* | *0.3* |
| Congo | 32 | **150 ± 90** | **100** | **50–100** | **8–5** | **0.9** |
| | | *170 ± 300* | *5* | *5* | *590* | *0.1* |
| Huanghe | 152 | **−860 ± 450** | **–** | **–** | **–** | **0.5** |
| | | *−60 ± 150* | *–* | *–* | *–* | *0.1* |

Results for GRACE s1 and s2 solutions are in bold and italic fonts, respectively. "Grain size" is the grain sizes that minimize the difference between the SSD and the in situ sediment discharge, for each river and each hydrological correction applied on GRACE solutions (only for rivers which are SSD positive). "Grain size range" gives the range of grain sizes for which the SSD does not differ by more than 10% from the best SSD (obtained for the best grain size). Narrow grain size ranges suggest that GRACE could help constraining the grain size of the particles that are responsible for the largest mass accumulation out of each river. The entire range of grain sizes tested in this work is 5–100 µm. The area of the sedimentation zone associated to the grain size range is also given. The last column is the SNR for each river (see also Methods)

advection model can deal with a combination of grain size, the lack of grain size data for most rivers lead us to work with a single and constant grain size to define sediment deposition areas.

The simulation runs until the particle reaches the sea bottom or passes 200 m depth. Each river has thus a set of eight simulated sedimentation zones, one per grain size (Supplementary Fig. 6).

Our modeling neglects the influence of tidal currents, which increases the computational time of the simulation without significantly altering the resulting sedimentation zones, at least at the spatio-temporal scale of this study. This is shown in Supplementary Fig. 7, where we compare the sedimentation zone computed only considering sea currents (described above) with the sedimentation zone computed considering both, sea currents and tidal currents. This comparison is done for the Amazon estuary, considering the strong tidal currents acting in this region. We use tidal currents from FES2014 products, provided by AVISO, and only consider the M2 tidal component, which has the highest amplitude there. This tidal velocity is added to the sea current (tidal currents are considered constant with depth). The magenta box in Supplementary Fig. 7 shows the area that concentrates most of the deposited sediments. This area contains 97.2% of the total exported mass neglecting tidal currents and 96.7% when considering them, hence a negligible difference of 0.5% which actually matters at the pixel scale (0.25°, so ~25 km sides) but drives to negligible difference at the scale of hundreds of kms as in our study. In addition to tides, waves can also participate in the sediment redistribution. We state that this effect remains negligible at our spatial and temporal scales, since waves-induced sediment redistribution is localized to the shore and culminates during episodic storms[48–50]. At geological time scales, though, wave-induced sediment redistribution might become a first-order process compared to advection by sea currents[51]. Also, we neglect bedload transport processes, which are responsible for delta front progradation. Indeed, examples of fast prograding systems exhibit velocities of, e.g., 5 km in 40 years at the Red River[52] or 10 m year⁻¹ for the Ganges-Brahmaputra[53]. Such velocities may significantly redistribute sediment at geological scales but not over our 10-year simulations of sediment transport. As a result, the change of mass, hence the gravity change, in this area is 0 and has therefore no effect on GRACE measurements.

Once computed, each sedimentation zone is then normalized by its total mass to get the distribution of the sediment mass over this zone. It is then spatially filtered to represent this distribution as if it was observed by GRACE, specifically

taking the same parameters as those constraining the GRACE solutions we use: the spatial distribution of the mass of sediment is first converted into spherical harmonics (SH) coefficients until 120° and then reconstructed in the spatial domain with a 0.5° resolution. Note that we assume that the volume of sediment that accumulates on the seafloor replaces the same volume of seawater, so that the actual gain of mass is the residual of the mass of sediment minus the mass of water displaced by the sediment particles.

**GRACE regularized solutions.** The GRACE regularized solutions used in this study were produced at the Center for Space Research (CSR). While the procedure and the technique to compute these solutions is still the same as described in ref.[20], these solutions have been reprocessed with improved parameters. Indeed, the error characterization to compute these regularized solutions has been significantly upgraded. The new procedure includes a robust estimation of error contribution from each spherical harmonic coefficient to the monthly GRACE solutions. This error characterization is then used to define the regularization matrix when computing the regularized GRACE solutions. GRACE processing also considers substitution of degree-1[54] and degree-2 coefficients[55].

A new set of GRACE mascon products are now available for use from centers like CSR[56] and JPL[57]. However, these solutions apply constraints based on land and ocean boundaries. We believe that these products are not suitable for this study because of the likelihood that these mascon solutions will attenuate the offshore sedimentation signals. Hence, we chose to use the regularized spherical harmonic solutions, instead of the mascon solutions, which do not have the land/ocean boundary constraints.

With significant improvement in the understanding of the GRACE system and errors, due to the improvement in the processing techniques over the years and with the upcoming GRACE RL06 reprocessing, it is expected that the SNR of the GRACE regularized solutions will be further improved in the upcoming release.

**Hydrological models.** Although we are focusing on oceanic areas, sedimentation zones generally begin near the river mouth and coastal regions. GRACE is spectrally limited, which generates spatial leakage of continental water masses into the coastal ocean, affecting sedimentation signal. Correction of hydrological signals at a global scale is based on an ensemble of land-surface and global hydrological models (Supplementary Table 4). Difference in modeled long-term trend[58] is damped by

the ensemble approach[21] and the inclusion of a model assimilating GRACE on the continents[59]. Since there is no best hydrological model so far, we consider the hydrological uncertainty by using two GRACE solutions s1 and s2, built on two different hydrological corrections. GRACE s1 is computed by removing the mean of seven different hydrological models, namely: GLDAS models (CLM, MOSAIC, NOAH v2.7, VIC), WGHM v2.1 and v2.2b, AWRA-L (Supplementary Table 4). GRACE s2 is computed by removing the Australian Water Resource Assessment (AWRA in Supplementary Table 5), which is assimilating GRACE data[59]. This approach is interesting as it considers GRACE signal on the continents to improve the representation of continental water storage changes. See Supplementary Fig. 1 and Supplementary Table 5 for details about each model.

**GRACE data information content and inversion strategy**. The goal of this section is to investigate the data quality of different GRACE products and evaluate their ability to estimate localized mass changes close to the coast. The approach is led with both CSR regularized GRACE solutions[56] and CSR RL5 SH solutions[60] with different processing strategies. At the same time, we test the inversion strategy in a numerical experiment and a real test case. Lake Aswan is found as a favorable example, considering its limited size (~5000 km$^2$) and its location in the desert but close to the Red Sea. The whole numerical and data experiment is therefore set up on this region.

**A numerical experiment: inverting a known mass**. Here, we evaluate potential mass estimation errors linked to (1) the presence of noise with different SNR and (2) errors associated with a wrong a priori, with a focus on an error on the sedimentation area to better represent uncertainties in the sediment advection model. Three different processing strategies are tested, linked to usual GRACE products: (1) truncation at 120° with no further filtering, mimicking regularized solutions; (2) truncation at 96° plus 300-km Gaussian smoother, associated to classical SH products with standard processing strategy; and (3) truncation at 96° plus ddk-7 filter[61,62]. These three products offer different actual spatial resolution. All computations are carried out considering a 1/16th-degree grid resolution.

Following several papers[63,64], GRACE noise structure is estimated for each GRACE product in the ocean, at the same latitude as the Aswan Lake. The extracted noise structure is normalized to zero mean and variance unit to be added to the expected signal. In all cases, the worst noise situation is considered, where the maximum of residual noise intercepts the lake. Considering that GRACE filtering generates smooth fields, we define the SNR as:

$$\text{SNR} = \frac{\max(\text{true mass@GRACE resolution})}{3.\text{std}(\text{noise field})}. \quad (1)$$

In the analysis, we consider several SNR ratios ranging from 0.1 to 10.

Further, we also investigate the impact of an incorrect a priori mass extension to recover variations. For the specific case of sediment mass accumulations, the maximum mass accumulation is generally well constrained, while its area depends on the sediment grain size. Therefore, we considered potential uncertainties linked to an unknown area of accumulation. The ratio of the unknown area to the actual area ranges from 0.1 to 20.

**The no noise case**. As a first step (Supplementary Fig. 10), the inversion of a known mass on a right a priori position with no noise is tested. As shown in the figures above, the whole mass can be recovered, whatever processing strategy is considered. Residuals are negligible. Note that T96G300 colorbar is not the same as the other products, the spatial pattern is also much wider. This strong smoothing remove noise but also affects spatial resolution in a significant way.

**Wrong a priori area**. In a second step (Supplementary Fig. 11), we consider no noise but a wrong a priori. The mass distribution covers an area which is 20 times larger (~100,000 km$^2$) than the lake itself (~5000 km$^2$, left panel). For both T96G300 and T96DDK7 processing strategies, with limited spatial resolution, the a priori filtered as GRACE and the true mass filtered as GRACE shows the same patterns, resulting in negligible residuals. With higher spatial resolution, the impact of a 5000 km$^2$ and a 100,000 km$^2$ mass distribution have different spatial patterns, leading to residuals. The mass recovered when resolving the forward model is, respectively, 30, 6, and 12% larger for T120, T96G300, and T120DDK7 processing strategies. This is coherent with findings from ref. [12] and highlights that high resolution also comes with a higher sensitivity to local masses and mass estimates dependent on a priori information.

**Adding noise**. In this third step (Supplementary Fig. 12), we consider a correct mass distribution, but added noise structures determined in the ocean at the same latitude as the Aswan lake (figures on the extreme left panel). Note that noise pattern is chosen to be positive above lake Nasser. Here, a SNR of 1 is considered. Comparing the "forward model" panel with previous figures shows that the mass recovered absorbs part of the noise structure, resulting in higher masses, and residuals differing from the input noise structure. Mass overestimation is, respectively, 30, 50, and 60% for T120, T96G300, and T96DDK7. In any case, the ability

to accurately estimate masses is related to the right balance between noise attenuation and spatial resolution conservation.

**Combining wrong a priori and noise**. In this last step (Supplementary Fig. 13), the cumulative impact of noise (SNR = 1) and wrong a priori is combined. Again, masses are overestimated, respectively, by 80, 50, and 80% for T120, T96G300, and T96DDK7. We can conclude that mass estimation with high-resolution solutions are more affected by the a priori distribution of masses than by noise as noise structure might be more independent from the shape of interest. On the contrary, standard-resolution products are more affected by noise structures, especially when the SNR decreases.

**A synthesis on the numerical experiment**. To sum up (Supplementary Fig. 14), the ability of each processing strategy to recover accurately some mass depends on the orthogonality between noise structure and a priori mass distribution for all processing strategies, but also on the knowledge of the true mass distribution for the high-resolution products. Overall, the proposed inversion strategy with inclusion of a priori mass distribution is powerful in the sense it can separate noise and signal if both spatial structures are independent. Even in a non-favorable case, masses can be recovered by a factor of 2 even at low SNRs below 1. For a right area a priori, the regularized solutions are expected to be less sensitive to noise by a factor ~2, with mass overestimation of a factor 2 at a SNR ~=0.3, while standard processing overestimated masses by a factor 2 at a SNR ~=0.6.

Note that results are shown for a positive noise over the lake area, adding up to the lake mass variations. In reality, noise can add up or subtract to the true mass variations, so the amplitude of the recovered mass can be smaller or even change sign for the lowest SNR.

**A real test case—trend estimates**. Following the results of the numerical experiment, the goal of this section it to apply the inversion method on the 5000-km$^2$ Aswan lake, recover long-term mass variations from GRACE and compare results to satellite altimetry. Lake Aswan has been monitored by several satellite altimetry missions, from 1995 up to now, providing long-term level changes[65]. Over the GRACE time period, Aswan lake level has decreased at the rate of −0.177 m year$^{-1}$ after removing seasonal variations, which is equivalent to a mass decrease of −0.93 Gt year$^{-1}$. Note that a constant area is considered, therefore the mass variations might be overestimated if the lake area decreases when water level decreases. Furthermore, the Aswan Dam is designed so that water rising above 178 m a.s.l is naturally diverted toward the Toshka depression in the Western Desert of Egypt[66]. In such cases, surplus water creates lakes located in the NE of Aswan Lake, of which the area was maximum in 2002 and then decreased onward. Both lakes are ~180 km away from each other.

Such a context offers the opportunity to test the sensitivity of the inversion to unknown mass changes that would not have been predicted by some a priori mass distribution. Two different inversions were tested. A first one, where the Aswan lake is considered alone, a second one where both Toshka and Aswan lakes are inverted separately. Results are shown in Supplementary Figs. 15, 16 and Supplementary Table 5. In this region where SNR ~=0.5, a factor ~2.5 or ~2 between true mass and inverted mass (see Supplementary Fig. 14) can be expected for standard or regularized products, respectively. When Aswan lake is inverted alone, regularized CSR underestimate the expected mass changes by a factor ~2, while both CSR5G300 and CSR5DDK7 overestimate mass changes by a factor ~2.5 and ~4 (Supplementary Fig. 15). When both Aswan and Toshka lake are inverted together, the total mass for low-resolution products is equivalent to the value recovered when inverting the Aswan lake alone, but highly unstable for the Aswan lake alone, suggesting the inability to separate both mass changes. On contrary, regularized CSR provides with a similar mass change for Aswan in both cases (Supplementary Fig. 16 and summary in Supplementary Table 5), highlighting the ability to separate both masses based on their specific spatial distribution.

These results highlight that low-resolution products will be sensitive to unmodeled mass changes in a region >200 km around the a priori mass distribution. This might be problematic for application to sediment mass changes, considering the proximity of the continent. On the contrary, regularized GRACE data shows its wide potential to separate close masses and ensure the stability of estimated mass changes.

**Lake time series inversion**. The same inversion procedure can be applied to recover the time variability of the time series, and not only their long-term trends. This exercise is more difficult as the GRACE noise structures evolve over time, leading to potential larger noise. Therefore, the idea is to evaluate the sensitivity of the GRACE processing strategy to different kind of noise structures that has not been considered in the previous numerical analysis. In this inversion, no temporal constrains have been applied, we apply the same spatial constrains (1) on Lake Aswan only and (2) on both lake Aswan and Toshka lakes. Inverted time series are generally noisy for low-resolution products, with a large overestimation of mass variations, as underlined in the previous experiment and limited correlation (<0.5) with respect to satellite altimetry data. The noise is even more striking on the low-resolution products when another degree of freedom is added by the Toshka lake. On the contrary, the regularized CSR product provides with a smooth time series

and correlation larger than 0.85. As previously, the amplitude is underestimated, but within the uncertainty estimate based on the SNR (Supplementary Fig. 17).

High-resolution GRACE products offer the opportunity to better separate signal and noise based on their spatial variability. It is clear from this experiment that regularized CSR data is well suited to estimate localized mass changes when the mass position and extent are approximatively known.

**Evaluating the impact of continental leakage**. The estimation of sedimentary mass changes is challenging at the 10-year time scale, considering that the amplitude of water storage changes is ~10 times larger. A previous attempt by Schnitzer et al.[67] was focusing on a continental region, where sedimentary processes are distributed over a wide area. They underlined that uncertainties on continental water storage changes are affecting the ability to recover sedimentary mass changes. In this work, the approach consists in focusing in the ocean, where basin-scale eroded masses are concentrated over a small region in the ocean. Although sedimentary masses are not on the continents, they generally deposit along the sea shore, where uncertainties on water storage changes can propagate into the ocean by leakage. The Amazon basin is a perfect example to test the impact of leakage: (1) as a tropical basin with large storage changes, it is expected that model uncertainties are large; (2) sedimentary masses are transported North along the coast by oceanic currents.

The different hydrological model outputs (see Methods) were turned to GRACE resolution for each processing strategy. The trend was computed for all models and their standard deviation was taken as an indication of the uncertainties generated by continental water storage changes (Supplementary Fig. 18). Uncertainties are generally large, even at GRACE resolution. Uncertainties are especially concentrated along the coast and at the mouth of the Amazon River; they are much larger for high-resolution (T120) with respect to low-resolution strategy (T96G300), which is coherent with similar studies[21]. However, spatial leakage is much more pronounced for low-resolution products. Eventually, leakage is limited to, respectively, 5, 11, and 11 mm year$^{-1}$ for T120, T96G300, and T96DDK7, resulting in a mass uncertainty of 0.25, 0.5, and 0.5 Gt year$^{-1}$, respectively.

As a conclusion, leakage errors are twice smaller for high-resolution products than for standard GRACE products. One more time, regularized CSR is more suited to study sedimentary masses.

**Code availability**. The computer code that simulates sediment's advection from ECCO2 velocity fields is available from the corresponding author on reasonable request.

**Data availability**. The GRACE datasets analyzed during the current study are available from the corresponding author on reasonable request. ECCO2 datasets used in this study are available from ftp://ecco.jpl.nasa.gov/ECCO2/cube92_latlon_quart_90S90N/.

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

## Acknowledgements

M.M. is supported by a postdoctoral grant by Centre National d'Etudes Spatiales (CNES) and by the CRITEX project funded by the Agence Nationale de la Recherche (ANR-11-EQPX-0011). We acknowledge support by the EROQUAKE project funded by the Agence Nationale de le Recherche (ANR-14-CE33-0005). We also acknowledge support from Université Rennes 1. We thank Alexandre Guay for his insights on a preliminary version of this work. We thank Jean Braun, Saggy Cohen, and Martine Simoes's insightful discussions. Mette Bendixen and Stephen Darby have provided stimulating reviews.

## Author contributions

M.M., L.L. and P.S. conceived the study and prepared the manuscript. M.M. built the sediment advection model and performed the main analysis. L.L. defined GRACE processing and inversion strategy. A.C. and C.R. helped with the analysis of river sediment discharges. H.S. prepared the regularized GRACE solutions. J.-M.L. helped in the interpretation of GRACE data. All authors contributed to the final form of the article.

## Additional information

**Competing interests:** The authors declare no competing interests.

