## [Peer Review File · Nature Communications]

Reviewers' comments:

Reviewer #1 (Remarks to the Author):

Thank you for the opportunity to review this very interesting contribution. I have structured my report using the main questions posed in the 'guidelines to referees', which broadly cover issues of novelty, significance, and rigour as follows:

Q: What are the major claims of the paper?

A: The paper employs a new geodetic methodology – using satellite gravimetry (specifically GRACE mission retrievals) to estimate the mass of sediment being deposited in the immediate offshore regions of large rivers since 2002 – to estimate the mean rate of sediment flux (Qs) emanating from the world's large rivers. Based on a comparison of GRACE derived QS estimates for a selection of these large rivers (the Amazon, Ganges, Changjiang, Indus, Magdalena, Godavari and Mekong, and others), the paper claims that the new method can reliably quantify mean annual timescales Qs for these very large rivers, within reasonably uncertainty bounds, at least for the period of GRACE retrievals thus far (i.e., 2002-2017).

This is a very important claim for a variety of reasons. The flux of sediment from the continents to the oceans (which is disproportionately dominated by the sediment load from the world's largest rivers) is important for a variety of reasons in the earth and environmental sciences: for example, sediments are the vectors of nutrients (playing a role in biogeochemical cycling) and the flux of sediments reaching coastal environments (e.g., deltas) plays a first order control on relative sea-level rise rates in such locations. However, as detailed in the introduction to the paper existing estimates of Qs are challenging and, very likely, unreliable. This is not only because of the difficulty of standardising measurement protocols across different river basins in different national territories, but also because the monitoring process is expensive and difficult. For this reason available time-series tend to be sporadic, and often short. Globally hydrological measurement programmes are in decline at precisely the time when major environmental changes may be driving significant shifts in Qs to the coast – so there is a pressing need for new ways of reliably estimating Qs. Moreover estimating Qs delivery *to the ocean* (note my emphasis) is often fundamentally constrained by the reality that existing measurement nodes tend to be located at stable gauging stations located in the non-tidal regions – often many hundreds of kilometres from the coast. We know almost nothing about how much of the sediment load as measured at these stations is attenuated by in-channel and delta-plain deposition. For all these reasons a novel, independent, technique that promises to deliver new insight into the Qs reaching the coast is extremely exciting.

Q: Do you feel that the paper will influence thinking in the field?

A: In short, yes. For all the reasons highlighted above the analysis and measurement of Qs to the oceans is fundamentally important in a range of disciplines. The new method promises to deliver data that could over turn our understanding of these issues. This is one of a very few papers in recent years that has genuinely excited me as I read it. I am sure that scientists will be intrigued by the potential of the new method and seeking ways to incorporate it into their future work and thinking. As the authors indicate the potentially really exciting breakthrough would be in finessing the approach to enable variations in Qs to be elucidated at finer temporal resolution, which would allow much greater insight into governing process. Nevertheless this represents the first key step on that journey and I suspect that this paper could turn into a citation classic – notwithstanding some of the inevitable technical concerns raised below.

Q: Is the work convincing, and if not, what further evidence would be required to strengthen the conclusions?

A: In general I find that the methods are described in detail and they appear to be robust, albeit other reviewers with specific expertise in geodesy would need to comment on the details. From my perspective there are a couple of areas where additional clarification would be welcome.

The first - albeit it would be very difficult to evaluate this - concerns the potential circularity on validating the GRACE estimates of Q_s against estimates of Q_s derived from various sources, and which for the reasons cited have questionable reliability. Although the authors are careful to cite the sources used in the empirical data compilation, it may be appropriate to provide further details (perhaps extending the supplementary information) of the basis on which the empirical Q_s data are gathered, quality assured, etc so as to better constrain their uncertainties. For example, from my own experience of the Mekong River, the compilation provided by Walling is a 2008 paper which compiles data for the period 1960s-2002. That is, the data ends at precisely the point when the GRACE data begin and so they are not directly comparable (particularly given the point that the upstream environment is changing rapidly). Presumably the other data are subject to similar constraints - the data for the Fly, for example, are from a 2004 study and so on. I appreciate that the limitations of the empirical data are such that this is precisely why the GRACE data are so useful, but these limitations should be more clearly acknowledged. The uncertainties might also be further constrained by comparing the GRACE data not only with the empirical data but also with recent global modelling studies (e.g. see papers by Sagy Cohen et al) that may provide a further independent, contemporary, reference point (albeit again subject to its own limitations).

My second concern lies in the use of the sediment advection model. This step is necessary because the GRACE data will reveal only the proportion of sediment deposited within the sedimentation zone close to the outlet of each river. This volume will be smaller than the total flux exported from the river because some of the sediment exiting the river will be transported away from the sedimentation zone by tidal currents, wave action and (in some cases) the river plume itself may transport sediments many hundreds of kilometres offshore. In systems where the advection model does not properly represent these processes there would be the risk that the GRACE estimated mass of sediment is not properly corrected (by the advection model) - assuming I have correctly understood why and how the advection model is used. If I have understood this correctly then offshore depositional systems that are highly retentive should result in GRACE-derived Q_s values that converge more closely to the measured values. In contrast, offshore sedimentation zones that are less retentive would be more reliant on the advection model being correct in order to 'unbias' the GRACE data. It could be that considering the retention potential of each system could help understand how the errors evident in Figure 3, which in turn may help to increase the extent to which the work is convincing.

Professor Stephen Darby
Southampton, 19th January 2018

Reviewer #2 (Remarks to the Author):

The manuscript presents a study on the development of a new geodetic method to infer sediment discharge rates of large rivers to the ocean. The gravitational attraction generated when sediment is accumulated in the ocean is measurable with GRACE satellite imagery and give insight into sediment fluxes in global river deltas, is the general approach behind the study.

The scope of the study and the concept of using GRACE-data make this a very interesting and potentially important study to monitor sediment fluxes. Thus, the novelty and impact alone qualifies the manuscript for publication. However, the manuscript has several major problems which the authors must consider and address:

- The title doesn't fit the content of the paper; the authors are investigating sediment accumulation rather than sediment discharge or fluxes. A more appropriate title could be "Assessing the modern rates of sedimentation in the ocean using satellite gravimetry". The

confusing terminology makes the manuscript difficult to read.

- It is difficult to follow what kind of signal the authors are analyzing using the GRACE dataset. As such, the paper is not clear to read without reading other publications.

- The sediment model used in this study is probably depth-averaged. This is problematic, since most of the rivers and associated deltas cause stratified flow due to freshwater influx into the ocean. This stratification significantly influences the sedimentation rates in the near-shore delta area. The depth-average 'ocean-current' approach (line 65) is in my opinion only valid at distances further away (>100 km) from the delta mouths.

- The paper mainly deals with the sedimentation rates on the ocean floor in front of the deltas. There could be a major discrepancy between the 'river flux estimation' and the ocean floor sedimentation, since there is also the possibility of delta front progradation.

Please find below our replies to reviewer #1 (Stephen Darby) and reviewer #2. We thank Stephen Darby and an anonymous reviewer for their constructive comments. Comments are in black font; our replies are in blue font.

Reviewer #1, Stephen Darby (Remarks to the Author):

Thank you for the opportunity to review this very interesting contribution. I have structured my report using the main questions posed in the 'guidelines to referees', which broadly cover issues of novelty, significance, and rigour as follows:

Q: What are the major claims of the paper?

A: The paper employs a new geodetic methodology – using satellite gravimetry (specifically GRACE mission retrievals) to estimate the mass of sediment being deposited in the immediate offshore regions of large rivers since 2002 – to estimate the mean rate of sediment flux (Qs) emanating from the world's large rivers. Based on a comparison of GRACE derived QS estimates for a selection of these large rivers (the Amazon, Ganges, Changjiang, Indus, Magdalena, Godavari and Mekong, and others), the paper claims that the new method can reliably quantify mean annual timescales Qs for these very large rivers, within reasonably uncertainty bounds, at least for the period of GRACE retrievals thus far (i.e., 2002-2017).

This is a very important claim for a variety of reasons. The flux of sediment from the continents to the oceans (which is disproportionately dominated by the sediment load from the world's largest rivers) is important for a variety of reasons in the earth and environmental sciences: for example, sediments are the vectors of nutrients (playing a role in biogeochemical cycling) and the flux of sediments reaching coastal environments (e.g., deltas) plays a first order control on relative sea-level rise rates in such locations. However, as detailed in the introduction to the paper existing estimates of Qs are challenging and, very likely, unreliable. This is not only because of the difficulty of standardising measurement protocols across different river basins in different national territories, but also because the monitoring process is expensive and difficult. For this reason, available time-series tend to be sporadic, and often short. Globally hydrological measurement programmes are in decline at precisely the time when major environmental changes may be driving significant shifts in Qs to the coast – so there is a pressing need for new ways of reliably estimating Qs. Moreover, estimating Qs delivery *to the ocean* (note my emphasis) is often fundamentally constrained by the reality that existing measurement nodes tend to be located at stable gauging stations located in the non-tidal regions – often many hundreds of kilometres from the coast. We know almost nothing about how much of the sediment load as measured at these stations is attenuated by in-channel and delta-plain deposition. For all these reasons a novel, independent, technique that promises to deliver new insight into the Qs reaching the coast is extremely exciting.

Dear Dr. Darby, thank you for your enthusiasm and constructive comments.

Q: Do you feel that the paper will influence thinking in the field?

A: In short, yes. For all the reasons highlighted above the analysis and measurement of Qs to the oceans is fundamentally important in a range of disciplines. The new method promises to deliver data that could overturn our understanding of these issues. This is one of a very few papers in recent years that has genuinely excited me as I read it. I am sure that scientists will be intrigued by the potential of the new method and seeking ways to incorporate it into their future work and thinking. As the authors indicate the potentially really exciting breakthrough would be in finessing the approach to enable variations in Qs to be elucidated at finer temporal resolution, which would allow much greater insight into governing process. Nevertheless, this represents the first key step on that journey and I suspect that this paper could turn into a citation classic – notwithstanding some of the inevitable technical concerns raised below.

Q: Is the work convincing, and if not, what further evidence would be required to strengthen the conclusions?

A: In general, I find that the methods are described in detail and they appear to be robust, albeit other reviewers with specific expertise in geodesy would need to comment on the details. From my perspective there are a couple of areas where additional clarification would be welcome.

R1.1 The first - albeit it would be very difficult to evaluate this – concerns the potential circularity on validating the GRACE estimates of Qs against estimates of QS derived from various sources, and which for the reasons cited have questionable reliability. Although the authors are careful to cite the sources used in the empirical data compilation, it may be appropriate to provide further details (perhaps extending the supplementary information) of the basis on which the empirical Qs data are gathered, quality assured, etc so as to better constrain their uncertainties. For example, from my own experience of the Mekong River, the compilation provided by Walling is a 2008 paper which compiles data for the period 1960s-2002. That is, the data ends at precisely the point when the GRACE data begin and so they are not directly comparable (particularly given the point that the upstream environment is changing rapidly). Presumably the other data are subject to similar constraints – the data for the Fly, for example, are from a 2004 study and so on. I appreciate that the limitations of the empirical data are such that this is precisely why the GRACE data are so useful, but these limitations should be more clearly acknowledged. The uncertainties might also be further constrained by comparing the GRACE data not only with the empirical data but also with recent global modelling studies (e.g. see papers by Sagy Cohen et al) that may provide a further independent, contemporary, reference point (albeit again subject to its own limitations).

We agree on this circularity problem on validating our estimates using uncertain in situ data. As suggested, we now provide a table (new Table S1) that summarizes time periods and information on the sediment discharge quantification for every river used in this study. However, these further details cannot be used to numerically evaluate the uncertainty on in situ QS. Therefore, we take in situ QS estimates as first order constraints against which we can test the GRACE-derived QS. Note that the circularity problem is the reason why we also introduce another validity criterion, namely the consistency of GRACE-derived QS between the two hydrological models used in GRACE data processing. Indeed, estimates are more reliable when they converge toward the same value under various processing strategies. A far-end

option would be to make no comparison to in situ QS, hence relying only on the stability of GRACE-derived estimates for all grain sizes and processing strategies. In this case only Magdalena and Changjiang's SSDs would be acceptable. We now discuss this in lines 177-185 (documents with marks) or 147-155 (document without mark)

“Given their acknowledged uncertainty, in situ estimates are considered as first order constrains rather than accurate values. Under these conditions, valid SSD are proposed for four out of the thirteen studied rivers, including the Amazon (s1: 560 ± 90 Mt.yr⁻¹; s2: 610 ± 170 Mt.yr⁻¹), Changjiang (s1: 430 ± 100 Mt.yr⁻¹ ; s2: 330 ± 100 Mt.yr⁻¹), Indus (s1: 150 ± 70 Mt.yr⁻¹; s2: 350 ± 120 Mt.yr⁻¹) and Magdalena (s1: 460 ± 90 Mt.yr⁻¹; s2: 670 ± 90 Mt.yr⁻¹) rivers (Fig. 4 and Table 1). Indeed, these rivers have the highest sediment loads in the world, hence the highest contribution to the GRACE data. Further, for the Changjiang and Magdalena, grain sizes for each processing strategy are identical, supporting the robustness of the SSD estimates.”

Concerning the comparison of GRACE with modelling results, we contacted Dr. Sagy Cohen who obligingly ran the WBMsed model for estimating Qs at the outlet of the 13 rivers studied in this work and provided us with the results (Table R1 and Figure R1). Aside from the Amazon, which returned Qs = 5 Gt/yr (~5 times larger than the commonly used 1 Gt/yr value), these modelling results agree with in-situ data for the largest sediment delivery river (Ganges-Brahamaputra to Godavari). Then the agreement decreases for rivers with lower in situ Qs. This is similar to our conclusions when comparing Qs estimated using GRACE with in situ Qs. Therefore, WBMsed further puts in-situ observations in perspective. Modeled sediment discharge for Congo and Mekong rivers are much larger than in-situ estimates and closer to GRACE estimates.

While such results are encouraging, the variability among the different estimation approaches keeps wide. Further questions arise from the respective sources of uncertainties of such different approaches.

While, we would prefer focusing the paper on GRACE-based estimates, we acknowledge the similarities among satellite and modelling approaches: standardized methodology at global scale and defined time range. Perspectives are really exciting, GRACE and GRACE-FO can therefore become a tool to improve global modeling approaches. The following has been added on lines 244-250 (documents with marks) or 210-216 (document without mark)

“As an integrated sediment mass estimates, GRACE data complements in situ measurements by widening the range of spatial and temporal scales and provides a homogenous method to quantify these masses at global scale. Therefore, it is expected that GRACE-type satellite missions would deliver new insights into processes and mechanisms controlling sediment redistribution. Global sediment erosion and transport models²⁹, would benefit from this novel sediment mass observation, independent from rivers' flow information³⁰.”

River	In situ Mt/yr	WBMsed Mt/yr	SSD Mt/yr
Amazon	778	4950	560 ± 90

			610±165
Ganges Brahmaputra	1081	765	1140±200 -890±750
Changjiang	477	396	430±100 330±100
Mississippi	51	163	-130±310 -190±380
Irrawaddy	333	365	-1100±610 50±270
Indus	206	61	150±70 350±120
Orinoco	76	970	-250±450 -110±240
Godavari	119	89	230±90 650±120
Mekong	75	565	580±150 80±110
Magdalena	150	130	460±90 670±90
Fly	111	138	-1980±660 -570±470
Congo	32	1440	150±90 170±300
Huanghe	152	15	-860±450 -60±150

Table R1 : Comparison of Qs as estimated from in situ measurements, WBMsed model and GRACE (SSD) data (GRACE data have two values per river depending on which hydrological correction is used).

Figure R1: Same as Figure 4 in the manuscript but we add the WBMsed Qs results in green on the same axis as the Grace-derived Qs.

R1.2 My second concern lies in the use of the sediment advection model. This step is necessary because the GRACE data will reveal only the proportion of sediment deposited within the sedimentation zone close to the outlet of each river. This volume will be smaller than the total flux exported from the river because some of the sediment exiting the river will be transported away from the sedimentation zone by tidal currents, wave action and (in some cases) the river plume itself may transport sediments many hundreds of kilometres offshore. In systems where the advection model does not properly represent these processes there would be the risk that the GRACE estimated mass of sediment is not properly corrected (by the advection model) – assuming I have correctly understood why and how the advection model is used. If I have understood this correctly then offshore depositional systems that are highly retentive should result in GRACE-derived Qs values that converge more closely to the measured values. In contrast, offshore sedimentation zones that are less retentive would be more reliant on the advection model being correct in order

to ‘unbias’ the GRACE data. It could be that considering the retention potential of each system could help understand how the errors evident in Figure 3, which in turn may help to increase the extent to which the work is convincing.

Indeed, our sedimentation model consists only in sediment particle advection by sea current, as inferred from the ECCO2 3-daily sea current velocities. Therefore, we have tested the effect of tidal currents on the estimated sedimentation zone, which estimation is the main purpose of our sediment advection model (see Fig. R2). This test is done for the Amazon estuary, because it has strong tidal currents. We use the tidal currents from FES2014 products, provided by AVISO, and only consider the M2 tidal component, which has the highest amplitude there. This tidal velocity, which is a function of latitude and longitude but independent of the depth, is added to the sea current.

Figure R2: Comparison of the effect of tidal currents on the estimated sedimentation zone. Assume A is the spatial deposition only considering ECCO2 currents and B the spatial deposition considering both ECCO2 and tidal currents, then this map corresponds to $100(B - A)/A$.

For of our study, the main conclusion is that tidal currents do not alter significantly the spatial distribution of the sedimentation zones relative to the spatial and temporal scales investigated using GRACE data. The magenta box shows the area that concentrates most of the deposit sediments. It corresponds to 97.2% of the total exported mass neglecting tidal currents and 96.7% when considering them, hence a negligible difference of 0.5%.

Waves can increase the bottom shear stress and resuspend sediment. Once resuspended, currents may again advect the sediment before they settle. We state that this effect remains negligible at our spatial and time scales, since waves-induced sediment redistribution is localized to the shore and culminates during episodic storms (Dufois et al. 2014, 2008; Ferré et al. 2008).

We summarize this discussion in the manuscript lines 518-535 (documents with marks) or 478-495 (document without mark)

“Our modelling neglects the influence of tidal currents, which increases the computational time of the simulation without significantly altering the resulting sedimentation zones, at least at the spatio-temporal scale of this study. This is shown in Fig. S7, where we compare the sedimentation zone computed only considering sea currents (described above) with the sedimentation zone computed considering both, sea currents and tidal currents. This comparison is done for the Amazon estuary, considering the strong tidal currents acting in this region. We use tidal currents from FES2014 products, provided by AVISO, and only consider the M2 tidal component, which has the highest amplitude there. This tidal velocity is added to the sea current (tidal currents are considered constant with depth). The magenta box in Fig. S7 shows the area that concentrates most of the deposited sediments. This area focuses 97.2% of the total exported mass neglecting tidal currents and 96.7% when considering them, hence a negligible difference of 0.5% which actually matters at the pixel scale (0.25° , so ~ 25 km sides) but drives to negligible difference at the scale of hundreds of kms as in our study. In addition to tides, waves can also participate in the sediment redistribution. We state that this effect remains negligible at our spatial and temporal scales, since waves-induced sediment redistribution is localized to the shore and culminates during episodic storms⁴⁶⁻⁴⁸. At geological time-scales, though, wave-induced sediment redistribution might become a first-order process compared to advection by sea currents⁴⁹.”

Transportation through rivers plumes is a challenging question. We can safely assume that they are horizontally advected by sea currents in a way that is properly simulated by our advection model. For instance, we compare for the Amazon river our simulated advection pattern in space and time with images from Aqua Modis (Moderate Resolution Imaging Spectroradiometer) reflectance at 667 nm (R667), which is proportional to the concentration of sediments (Nechad et al. 2010). We do not know the calibration factor between R667 and the actual sediment concentration, but the normalized, adimensional, map remain insightful. We compute

1. the monthly relative deviation of the reflectance from the annual mean (using Modis images)
2. the monthly relative deviation of the sediment distribution from the annual mean (using our advection results).

and compare these monthly maps in Fig. R3. The general agreement between the spatial and temporal changes in reflectance/sediment patterns shows that our advection model correctly reproduces shallow plume migration.

The unknowns are 1) the sinking velocity of the plumes, which controls how far the plumes can go from the river mouth and 2) their ability to trigger turbidity currents, which will further redistribute sediment. Indeed, Hizzett et al. (2017) suggest that settling from surface river plumes may dominate the triggering of turbidity currents. But we cannot quantify these uncertainties.

Figure R3: Monthly difference in the sediment distribution (for the advection model, row 1 and 3 and for the Modis reflectance at 667 nm, rows 2 and 4) relative to the annual mean. Colorscale is qualitative, centered to 0, blue being negative and red positive. Note the coherent seasonal cycle in the spatial pattern between our advection model and Modis observations. The advection model was run for fine particles ($1 \mu\text{m}$ diameter), which remain at shallow depths due to their slow settling velocity.

References

- Dufois, François, Pierre Garreau, Pierre Le Hir, and Philippe Forget. 2008. "Wave- and Current-Induced Bottom Shear Stress Distribution in the Gulf of Lions." *Continental Shelf Research* 28 (15): 1920–34. doi:10.1016/j.csr.2008.03.028.
- Dufois, François, Romaric Verney, Pierre Le Hir, Franck Dumas, and Sabine Charmasson. 2014. "Impact of Winter Storms on Sediment Erosion in the Rhone River Prodelta and Fate of Sediment in the Gulf of Lions (North Western Mediterranean Sea)." *Continental Shelf Research* 72. Elsevier: 57–72. doi:10.1016/j.csr.2013.11.004.
- Ferré, B., X. Durrieu de Madron, C. Estournel, C. Ulses, and G. Le Corre. 2008. "Impact of Natural (Waves and Currents) and Anthropogenic (Trawl) Resuspension on the Export of Particulate Matter to the Open Ocean. Application to the Gulf of Lion (NW Mediterranean)." *Continental Shelf Research* 28 (15): 2071–91. doi:10.1016/j.csr.2008.02.002.

- L., Hizzett J, Hughes Clarke J E., Sumner E J., Cartigny M J B., Talling P J., and Clare M A. 2018. "Which Triggers Produce the Most Erosive, Frequent, and Longest Runout Turbidity Currents on Deltas?" *Geophysical Research Letters* 45 (2). Wiley-Blackwell: 855–63. doi:10.1002/2017GL075751.
- Nechad, B., K. G. Ruddick, and Y. Park. 2010. "Calibration and Validation of a Generic Multisensor Algorithm for Mapping of Total Suspended Matter in Turbid Waters." *Remote Sensing of Environment* 114 (4). Elsevier Inc.: 854–66. doi:10.1016/j.rse.2009.11.022.
- Salles, Tristan, Xuesong Ding, M W Jody, Ana Vila-concejo, Gilles Brocard, and Jodie Pall. 2018. "A Unified Framework for Modelling Sediment Fate from Source to Sink and Its Interactions with Reef Systems over Geological Times" 8 (December 2017): 1–11. doi:10.1038/s41598-018-23519-8.

Reviewer #2 (Remarks to the Author):

The manuscript presents a study on the development of a new geodetic method to infer sediment discharge rates of large rivers to the ocean. The gravitational attraction generated when sediment is accumulated in the ocean is measurable with GRACE satellite imagery and give insight into sediment fluxes in global river deltas, is the general approach behind the study.

The scope of the study and the concept of using GRACE-data make this a very interesting and potentially important study to monitor sediment fluxes. Thus, the novelty and impact alone qualify the manuscript for publication. However, the manuscript has several major problems which the authors must consider and address:

Dear reviewer, thank you very much for encouragements and questions. You will find below answers to your questions:

R2.1 The title doesn't fit the content of the paper; the authors are investigating sediment accumulation rather than sediment discharge or fluxes. A more appropriate title could be "Assessing the modern rates of sedimentation in the ocean using satellite gravimetry". The confusing terminology makes the manuscript difficult to read.

In our study, the total amount of sediment delivered by each river is deposited in offshore sedimentation zones. Indeed, GRACE satellite focuses on mass changes in sedimentation zones, but the analysis is driven by the sediment-advection model, which translates the month to month sediment discharge into sedimentation zones. Therefore, over the 10-years studied in this work, we consider that the sediment discharge and the sedimentation rate are identical.

We clarify this at line 83-85 (documents with marks) or 79-81 (document without mark):

"The resulting sedimentation zones (Figs. 2a-b-c and Fig. S3) have the same the same spatial resolution as ECCO2 (0.25 degree) and contain the total amount of sediment that was delivered at their respective rivers' outlets over the time of simulation (10 years)."

R2.2 It is difficult to follow what kind of signal the authors are analyzing using the GRACE dataset. As such, the paper is not clear to read without reading other publications.

We add this clarification at the first mention of GRACE in the article on lines 45-48

"GRACE measures gravity changes as a function of time and position over the whole earth, at monthly time scales. Gravity being physically linked to masses, GRACE data is fundamentally sensitive to the amplitude of mass variations, with no minimum threshold of spatial extension ¹²"

Details were also added on lines 120-129 (documents with marks) or 107-110 (document without mark)

"We analyse the linear gravity change computed from ten years (2002-2011) of monthly regularized GRACE solutions. They are originally provided in equivalent water thickness that we convert in equivalent sediment thickness assuming that sediment (density 2.5)

replaces water (density 1). ”

The methods (lines 553-574 (documents with marks) or 513-534 (document without mark)) and supplementary notes (starting at line 617 (documents with marks) or 563 (document without mark)) also provide more details on the GRACE data.

R2.3 The sediment model used in this study is probably depth-averaged. This is problematic since most of the rivers and associated deltas cause stratified flow due to freshwater influx into the ocean. This stratification significantly influences the sedimentation rates in the near-shore delta area. The depth-average ‘ocean-current’ approach (line 65) is in my opinion only valid at distances further away (>100 km) from the delta mouths.

Indeed, the vertical heterogeneity is important to take into account. We emphasize that our oceanic sediment advection model uses the ECCO2 model, which provides a 3D velocity field (zonal, meridional and vertical) of oceanic currents. It is therefore not a depth-averaged model as specified in lines 77-82 (documents with marks) or 73-78 (document without mark).

“The advection model combines the monthly sediment discharge of each river (see Fig. S2) with time-variable oceanic 3D currents velocities produced by the project Estimating the Circulation and Climate of the Ocean, Phase II (ECCO2¹⁹, see Method and Table S3), which horizontal resolution is 0.25 degree, vertical resolution ranges from 10 to 450 m (resolution decreases with depth) and temporal resolution is 3 days. This model tracks sinking sediment particles in accordance with the dynamics of both the sediment discharge and the currents.”

The depth of the sediment is re-evaluated at each iteration to apply the appropriate depth-dependent velocity and simulate its trajectory. We yet agree that the spatial resolution of the oceanic current model (0.25 degrees horizontally and from 10 to 450 in depth – the resolution gets lower at higher depths) is probably too coarse to offer realistic sedimentation patterns in the near-shore delta area. However, our sediment advection model only aims at defining the spatial extent of the sedimentation zone and the area where most of the sediment concentrates, which is mainly on the shore near the river mouth (~100 km).

R2.4 The paper mainly deals with the sedimentation rates on the ocean floor in front on the deltas. There could be a major discrepancy between the ‘river flux estimation’ and the ocean floor sedimentation, since there is also the possibility of delta front progradation.

We evaluated this issue by looking at examples of fast prograding systems. They exhibit velocities of, e.g., 5 km in 40 years at the Red River (Van Maren 2005) or 10 m/yr for the Ganges-Brahmaputra (Swenson et al. 2005). Such velocities may significantly redistribute sediment at geological scales but not over our 10-years simulations of sediment transport. As a result, the change of mass, hence the gravity change, in this area is 0 and has therefore no effect on GRACE measurements.

This is clarified on lines 535-541 (documents with marks) or 495-501 (document without mark).

“Also, we neglect bedload transport processes, which are responsible for delta front progradation. Indeed, examples of fast prograding systems exhibit velocities of, e.g., 5 km

in 40 years at the Red River 50 or 10 m/yr for the Ganges-Brahmaputra 51. Such velocities may significantly redistribute sediment at geological scales but not over our 10-years simulations of sediment transport. As a result, the change of mass, hence the gravity change, in this area is 0 and has therefore no effect on GRACE measurements.”

References

Maren, D S van. 2005. “Barrier Formation on an Actively Prograding Delta System: The Red River Delta, Vietnam.” *Marine Geology* 224 (1): 123–43.
doi:<https://doi.org/10.1016/j.margeo.2005.07.008>.

B., Swenson John, Paola Chris, Pratson Lincoln, Voller Vaughan R., and Murray A Brad. 2005. “Fluvial and Marine Controls on Combined Subaerial and Subaqueous Delta Progradation: Morphodynamic Modeling of Compound-clinof orm Development.” *Journal of Geophysical Research: Earth Surface* 110 (F2). Wiley-Blackwell.
doi:[10.1029/2004JF000265](https://doi.org/10.1029/2004JF000265).

REVIEWERS' COMMENTS:

Reviewer #1 (Remarks to the Author):

The authors have provided a substantially revised version of the original manuscript and included a detailed, reasoned, responses to reviewers document. I am completely satisfied that the issues raised in my previous review have now been addressed. To be completely clear on this point, I am satisfied that the new revisions have further strengthened the methodological rigour of the paper to the point where it is clear that this paper is robust as well as being original and important.

In reading the revised paper I have noticed a few passages of text where there are some very minor grammatical errors, typographical errors, or phrasing that could be edited for clarity. In preparing the paper for publication it would be helpful to consider these as follows:

- L34 - suggest 'is' not 'are'
- L50 - suggest rephrasing to 'sediment fluxes delivered from major rivers'
- L58 - suggest 'as in the case of the Amazon and....'
- L64 - suggest '(the GRACE mission lasted....)'
- L74 - suggest 'current' not 'currents'
- L79 - delete repetition of 'the same'
- L86 - delete 'solely'
- L109 - suggest 'convert to' rather than 'convert in'
- L146 - suggest 'reduce the difference with the in situ estimates'
- L148 - use 'constraints' not 'constrains'
- L165 - suggest 'even if the SSD fails to...'
- L203 - suggest 'directly measure suspended....'
- L207 - suggest 'with the exception of'
- L209 - suggest 'will pave the way to achieving better'
- L211 - use 'complement' not 'complements'
- L212 - use 'providing' not 'provides'
- L213 - suggest 'the global scale'

Reviewer #2 (Remarks to the Author):

Dear authors and Editor,

I have read the manuscript and I value the thorough job the authors have done to answer our questions and concerns, and I am happy to see they've taken many of our suggestions into considerations and changed accordingly. Particularly, I am glad to see that more details on methodology have been added to the main text. This makes the manuscript easier to read.

I do still argue that the authors should be careful with the use of the word "sediment discharge". What they develop is a method to infer the accumulation of sediment concentrations in the offshore zone. As the authors state in the introduction, measurements of suspended and bedload sediment come with several challenges. The proposed method does not determine the fluxes of suspended and bedload sediment, but focus on determining the endmember of the process of sediment transport from land to sea; the accumulation of sediment in the ocean. I suggest using "sediment mass accumulation" or "sediment load" when referring to "sediment discharge" as well as "SSD" and change this throughout the text. Moreover, in line 204-205 (discussion), I suggest writing the sentence as: [... this novel method measures sediment fluxes derived from continuous mass accumulation in the ocean].

Furthermore, it is my opinion, that the authors should not oversell the applicability of the method – as it is written now, the applicability is overblown. In the abstract (line 16) and in the discussion, the authors state that the approach opens a new era for monitoring large rivers – “large” is a rather arbitrary definition and I suggest being more specific by stating “the world’s largest rivers”. Since the resolution of GRACE is 170 km, signals will only be observed at the largest rivers on Earth.

Lastly, the discussion is skewed towards too much focus on perspectives and expectations for future missions, rather than a discussion on the limitations on the method presented in the manuscript.

I still have a few questions, that needs to be addressed:

Line 20: Why is Godavari and Mekong no longer presented in the abstract?

Line 21: “worldwide erosion”. Please specify what type of erosion are you referring to? Coastal? Continental?

Line 79: Check for typo: “the same the same”

In general:

You state that the high-resolution data is 170 km – I was under the impression the GRACE satellite obtained data at a much higher resolution. Could you please clarify the 170 km resolution?

Yours sincerely,
Mette Bendixen

Reply to the reviewers

Dear Stephen Darby and Mette Bendixen,

Thank you for your comments and corrections. Please find below our reply in blue font, while reviewer's comments are in black font.

Maxime Mouyen and co-authors

Reviewer #1 (Remarks to the Author):

The authors have provided a substantially revised version of the original manuscript and included a detailed, reasoned, responses to reviewers document. I am completely satisfied that the issues raised in my previous review have now been addressed. To be completely clear on this point, I am satisfied that the new revisions have further strengthened the methodological rigour of the paper to the point where it is clear that this paper is robust as well as being original and important.

Thank you very much for your constructive and encouraging comments. We are happy that our last reply satisfied you.

In reading the revised paper I have noticed a few passages of text where there are some very minor grammatical errors, typographical errors, or phrasing that could be edited for clarity. In preparing the paper for publication it would be helpful to consider these as follows:

L34 - suggest 'is' not 'are'

L50 - suggest rephrasing to 'sediment fluxes delivered from major rivers'

L58 - suggest 'as in the case of the Amazon and....'

L64 - suggest '(the GRACE mission lasted....)'

L74 - suggest 'current' not 'currents'

L79 - delete repetition of 'the same'

L86 - delete 'solely'

L109 - suggest 'convert to' rather than 'convert in'

L146 - suggest 'reduce the difference with the in situ estimates'

L148 - use 'constraints' not 'constrains'

L165 - suggest 'even if the SSD fails to...'

L203 - suggest 'directly measure suspended....'

L207 - suggest 'with the exception of'

L209 - suggest 'will pave the way to achieving better'

L211 - use 'complement' not 'complements'

L212 - use 'providing' not 'provides'

L213 - suggest 'the global scale'

These corrections were all applied in the manuscript, with "R1" as a comment for easier tracking.

Reviewer #2 (Remarks to the Author):

Dear authors and Editor,

I have read the manuscript and I value the thorough job the authors have done to answer our questions and concerns, and I am happy to see they've taken many of our suggestions into considerations and changed accordingly. Particularly, I am glad to see that more details on methodology have been added to the main text. This makes the manuscript easier to read.

Thank you for your constructive comments and corrections. We are happy that our last reply and corrections satisfied you and we hope that this new version will satisfy you as well.

R2.1 I do still argue that the authors should be careful with the use of the word “sediment discharge”. What they develop is a method to infer the accumulation of sediment concentrations in the offshore zone. As the authors state in the introduction, measurements of suspended and bedload sediment come with several challenges. The proposed method does not determine the fluxes of suspended and bedload sediment but focus on determining the endmember of the process of sediment transport from land to sea; the accumulation of sediment in the ocean. I suggest using “sediment mass accumulation” or “sediment load” when referring to “sediment discharge” as well as “SSD” and change this throughout the text. Moreover, in line 204-205 (discussion), I suggest writing the sentence as: [... this novel method measures sediment fluxes derived from continuous mass accumulation in the ocean].

“Sediment discharge” and “sediment load” are considered as equivalent terms, as for instance stated, page 24, by Milliman and Farnsworth (2011): “*Sediment discharge, often termed sediment load, refers to sediment transport: mass per unit time.*”. We acknowledge that “sediment load” is commonly used among geomorphologists. Yet, because Nature Communications is a multidisciplinary journal, we prefer to use the word discharge than load, which in many disciplines is considered as a force. We believe that avoiding this possible misunderstanding will help the paper to have a broader impact.

Also, our hypothesis, now stated more clearly in the Introduction, is that the mass of sediment accumulating at the mouth of a river over a given time t is the sediment discharge of that river multiplied by t .

See **lines 51-53**:

“Our main hypothesis is that the mass of sediment exiting a river and accumulating in the ocean over a given time t is the sediment discharge of that river multiplied by t .”

R2.2 Furthermore, it is my opinion, that the authors should not oversell the applicability of the method – as it is written now, the applicability is overblown. In the abstract (line 16) and in the discussion, the authors state that the approach opens a new era for monitoring large rivers – “large” is a rather arbitrary definition and I suggest being more specific by stating “the world’s largest rivers”. Since the resolution of GRACE is 170 km, signals will only be observed at the largest rivers on Earth.

We rewrote according to your comment.

Abstract: modification on **lines 15-16**:

“Here, we develop a new geodetic method to infer the sediment discharge to the ocean of the world’s largest rivers.”

Discussion: modification on **lines 221-223**:

“Our approach brings new perspectives for the monitoring of continental erosion and rates of sediment mass accumulation out of rivers having the world’s largest sediment discharges using satellite gravimetry.”

We also modified other mentions of “large rivers” throughout the manuscript.

Please note that ability of GRACE to measure mass change signals does not depend on the size/extent of the mass change but on its amplitude (GRACE is sensitive to masses, not spatial scales). Indeed, on contrary to classical remote sensing tools where the resolution limits the interpretation, 170 km is a “gravimetric resolution”, which should be understood as an “integration domain”. This is shown in several papers, including Longuevergne et al. (2013) showing that GRACE data can be interpreted as mass changes over regions as small as 5000 km².

R2.3 Lastly, the discussion is skewed towards too much focus on perspectives and expectations for future missions, rather than a discussion on the limitations on the method presented in the manuscript.

We now discuss the limitation of our method by elaborating on 3 points: the need for grain-size data to improve our sedimentation zone estimates, the limitation due to GRACE leakage effect (also developed in the Method) and, as a result, our focus on rivers with the world’s largest sediment discharge. See **lines 239-248**:

“Yet the grain size significantly controls the extension of the estimated sedimentation zone. We thus tested a range of possible grain sizes, but robust estimates of the grain size distribution compiled in accessible database would make our sedimentation zones estimates more reliable. Another limitation comes from the GRACE data, which inherently suffer from leakage of hydrological signals into the coastal ocean²¹. Although such a leakage effect is efficiently reduced in the regularized solutions we use (see Method), it prevents from identifying masses of sediment accumulated out of rivers with sediment discharges smaller than ~150 Mt yr⁻¹ (Fig. 4 and Table 1) and makes this study focused on rivers having the world’s largest sediment discharges.”

I still have a few questions, that needs to be addressed:

R2.4 Line 20: Why is Godavari and Mekong no longer presented in the abstract?

We removed them from the abstract because, although they have plausible SSD, the difference between s1 and s2 for each of them is large, which does not fit our validation criteria (SSD values in agreement with in situ estimates and s1 and s2 lead to similar values). We add the missing explanation on **lines 203-206**:

“Godavari and Mekong river’s SSD are also significantly modified by removing Sumatra-Nias’ coseismic effect (Supplementary Table 2) and brought closer to in situ estimates, yet s_1 and s_2 solutions are not similar (factor 3 and 7 for Godavari and Mekong, respectively)”.

R2.5 Line 21: “worldwide erosion”. Please specify what type of erosion are you referring to? Coastal? Continental?

We now specify continental erosion, on **lines 20-22**:

“This new approach enables to quantitatively monitor the contemporary erosion of continental basins drained by rivers with large sediment discharges...”

R2.6 Line 79: Check for typo: “the same the same”

Corrected.

R2.7 In general: You state that the high-resolution data is 170 km – I was under the impression the GRACE satellite obtained data at a much higher resolution. Could you please clarify the 170 km resolution?

In fact, standard GRACE solutions are usually provided at a lower resolution, with spherical harmonics (SH) degree up to $l=60$, the horizontal scale being $\sim 20000/l$, this corresponds to ~ 330 km resolution in the spatial domain. Moreover, these standard GRACE solutions also have to be filtered, lowering again the spatial resolution to ~ 500 km (Wahr et al. 2004; Swenson and Wahr 2006). The solutions we used were computed up to SH degree 120, hence a spatial resolution of $20000/120 \sim 170$ km and no further filtering was needed (Save, Bettadpur, and Tapley 2012). We clarify why we write 170 km resolution on **lines 146-149**:

“These spherical harmonic solutions are computed up to degree and order 120 (see Method), which corresponds to a spatial resolution of ~ 170 km. Note that GRACE is sensitive to masses, not spatial scales¹². The 170-km resolution should be understood as an integration domain rather than a pixel-size information.”

References

- Longuevergne, L., C. R. Wilson, B. R. Scanlon, and J. F. Crétaux. 2013. “GRACE Water Storage Estimates for the Middle East and Other Regions with Significant Reservoir and Lake Storage.” *Hydrology and Earth System Sciences* 17 (12): 4817–30. doi:10.5194/hess-17-4817-2013.
- Milliman, John D., and Katherine L. Farnsworth. 2011. *River Discharge to the Coastal Ocean: A Global Synthesis*. Cambridge University Press.
- Save, Himanshu, Srinivas Bettadpur, and Byron D. Tapley. 2012. “Reducing Errors in the

GRACE Gravity Solutions Using Regularization." *Journal of Geodesy* 86 (9): 695–711. doi:10.1007/s00190-012-0548-5.

Swenson, Sean, and John Wahr. 2006. "Post-Processing Removal of Correlated Errors in GRACE Data." *Geophysical Research Letters* 33 (8): 1–4. doi:10.1029/2005GL025285.

Wahr, John, Wahr John, Swenson Sean, Zlotnicki Victor, and Velicogna Isabella. 2004. "Time-Variable Gravity from GRACE: First Results." *Geophysical Research Letters* 31 (11). Wiley-Blackwell: L11501. doi:10.1029/2004GL019779.